# A Systematic Review of Intracellular Microorganisms within *Acanthamoeba* to Understand Potential Impact for Infection

**DOI:** 10.3390/pathogens10020225

**Published:** 2021-02-18

**Authors:** Binod Rayamajhee, Dinesh Subedi, Hari Kumar Peguda, Mark Duncan Willcox, Fiona L. Henriquez, Nicole Carnt

**Affiliations:** 1School of Optometry and Vision Science, University of New South Wales (UNSW), Sydney, NSW 2052, Australia; h.peguda@unsw.edu.au (H.K.P.); m.willcox@unsw.edu.au (M.D.W.); n.carnt@unsw.edu.au (N.C.); 2Department of Infection and Immunology, Kathmandu Research Institute for Biological Sciences (KRIBS), Lalitpur 44700, Nepal; 3School of Biological Sciences, Monash University, Clayton, VIC 3800, Australia; dinesh.subedi@monash.edu; 4Institute of Biomedical and Environmental Health Research, School of Health and Life Sciences, University of the West of Scotland (UWS), Paisley PA1 2BE, UK; fiona.henriquez@uws.ac.uk

**Keywords:** *Acanthamoeba*, intracellular microbes, survival, carrier, co-infection

## Abstract

*Acanthamoeba*, an opportunistic pathogen is known to cause an infection of the cornea, central nervous system, and skin. *Acanthamoeba* feeds different microorganisms, including potentially pathogenic prokaryotes; some of microbes have developed ways of surviving intracellularly and this may mean that *Acanthamoeba* acts as incubator of important pathogens. A systematic review of the literature was performed in order to capture a comprehensive picture of the variety of microbial species identified within *Acanthamoeba* following the Preferred Reporting Items for Systematic Reviews and Meta-Analysis (PRISMA) guidelines. Forty-three studies met the inclusion criteria, 26 studies (60.5%) examined environmental samples, eight (18.6%) studies examined clinical specimens, and another nine (20.9%) studies analysed both types of samples. Polymerase chain reaction (PCR) followed by gene sequencing was the most common technique used to identify the intracellular microorganisms. Important pathogenic bacteria, such as *E. coli*, *Mycobacterium* spp. and *P. aeruginosa,* were observed in clinical isolates of *Acanthamoeba*, whereas *Legionella*, adenovirus, mimivirus, and unidentified bacteria (*Candidatus*) were often identified in environmental *Acanthamoeba*. Increasing resistance of *Acanthamoeba* associated intracellular pathogens to antimicrobials is an increased risk to public health. Molecular-based future studies are needed in order to assess the microbiome residing in *Acanthamoeba*, as a research on the hypotheses that intracellular microbes can affect the pathogenicity of *Acanthamoeba* infections.

## 1. Introduction

*Acanthamoeba*, a ubiquitously distributed free-living amoeba, is known to cause a rare, but potentially sight-threatening, painful, often misdiagnosed, and difficult to treat corneal infection, keratitis, and meningoencephalitis, a fatal infection of the central nervous system (CNS) [1,2,3,4,5]. *Acanthamoeba* spp. can also cause sinusitis and cutaneous lesions in immunocompromised individuals, such as AIDS patients [3,4,6]. It has two distinct stages in its life cycle, an active phagotrophic trophozoite and a quiescent double walled cyst stage, with the cyst stage enabling the amoeba to remain viable for many years, even in harsh conditions, including chlorine treated water [7,8]. The infective form is the trophozoite stage, although both trophozoites and cysts can gain entry into the human body via different routes, such as debrided skin, cornea, and nasal passages [9]. Based on their morphology, *Acanthamoeba* species have been broadly classified into three groups (I, II, and III) [10] and pathogenic strains are common of group II [11]. *Acanthamoeba* species are also classified into at least 22 (T1–T22) genotypes based on their 18S rRNA sequences, with species, such as *A. castellanii* and *A. polyphaga,* within the T4 genotype frequently associated with corneal infection [12,13,14].

The *Acanthamoeba* trophozoite feeds on other microbes, such as bacteria, fungi, algae, and viruses, and can carry them intracellularly acting as “Trojan horse” [15,16]. Therefore, *Acanthamoeba* can act as a vector of potentially pathogenic microorganisms and, hence, play a role in pathogen dissemination as well as acting as a pathogen itself [17,18,19]. Both clinical and environmental isolates of *Acanthamoeba* harbour pathogenic prokaryotes as endosymbionts [20,21,22]. The term “endosymbiont” has been described as “a regulated, harmonious cohabitation of two nonrelated partners, in which one of them lives in the body of the other”, and a bacterium is considered to be an endosymbiont when it is able to institute a replicative niche within, for example, eukaryotic cells [23]. However, another generic term “endocytobiont” has been coined to name the intracellular microbes that are associated with free-living amoeba to overcome any suggestion that the intracellular microbes might show mutualism, symbiosis, parasitism, phoresy, or zoochory [24,25]. Throughout the remainder of this review, the term “intracellular” will be used to encompass endosymbionts, endocytobionts, and other forms intracellular microbes within *Acanthamoeba* spp.

The detailed molecular pathways and strategies of intracellular interactions between *Acanthamoeba* and other microbes are unexplored at present. In a more generalised context, *Acanthamoeba* shares similar morphological and ultrastructural features to macrophages and they have a similar mechanism of interaction with microbes [26]. Amoeba may possess universal classes of receptors which bind with a wide array of microbial receptors facilitating adhesion and engulfment of a diverse range of microbes, such as Gal/GalNAc on *Legionella pneumophila* [27] or type III secretion structures on *Vibrio parahaemolyticus* [28] and *E. coli* K1 [29,30] (Figure 1). If the engulfed microbes can then escape the normal phagosome-associated feeding pathway, they may exist intracellular [18]. The ability of microbes to set up an intracellular lifestyle in *Acanthamoeba* and remain viable has been hypothesised to allow them to subsequently live intracellularly in mammalian cells [31,32]. The intracellular survival mechanisms of bacteria in the amoebal cytoplasm differ between species and this, coupled with analysis of phylogenetic lineages of intracellular bacteria, indicates the ability that has developed with time over the microbe’s evolution [33]. For instance, *V. cholerae* can escape degradation by applying an intricate neutralising program that effectively neutralises changes to the pH, digestive enzyme functions, and the production of reactive oxygen radicals that may otherwise destroy the bacteria [34]. On the other hand, *L. pneumophila* forms a membrane-enclosed microenvironment within the *Acanthamoeba* that is produced via the endoplasmic reticulum, membrane transporters, and fusion with other membrane-bound vesicles [35,36]. The intracellular survival and proliferation of bacteria in amoebal cells has been associated with enhanced resistance of bacteria to antimicrobials and increased bacterial pathogenicity [37]. *Acanthamoeba* containing intracellular bacteria, such as *Pseudomonas*, *Mycobacterium,* and *Chlamydia,* has demonstrated a more rapid cytopathic effect (CPE) in in vitro as compared to isolates without intracellular bacteria [21,38], showing enhanced amoebal pathogenic potential.

This systematic review examines the intracellular microorganisms in *Acanthamoeba* and compares the types of microbial species that were identified in environmental and clinical isolates of *Acanthamoeba*, and potential impact of intracellular microorganisms on *Acanthamoeba* keratitis. The major aims of this review are: (a) to determine the laboratory techniques that have been used for the isolation and identification of intracellular microbes in *Acanthamoeba* spp.; (b) to assess whether different ways of culturing *Acanthamoeba* affect the types of intracellular bacteria; (c) to examine which microbes are most commonly found inside *Acanthamoeba* spp.; and (d) to determine whether environmental and clinical isolates of *Acanthamoeba* harbor the same intracellular prokaryotes.

## 2. Results

### 2.1. Results of the Search

The electronic search identified 1331 articles (PubMed = 234, Scopus = 704, WoS = 393). After the removal of duplicates (*n* = 138), 1193 articles were screened based on their titles and abstracts. The outcome was that 43 studies met the inclusion criteria. Figure 2 depicts the screening process.

### 2.2. Included Studies

In total, 43 studies were analysed. The study location, sample type, laboratory methods used, species and genotypes of *Acanthamoeba* strains, types of intracellular microbes, and co-occurrence of multiple microorganisms were examined. Brief details of each study included in the analysis are mentioned in Table 1.

### 2.3. Laboratory Techniques Used for the Isolation and Identification of Intracellular Microbes in Acanthamoeba spp.

Microbial culture, fluorescence in situ hybridization (FISH), microscopy, polymerase chain reaction (PCR), gene sequencing, and gas liquid chromatography were the laboratory techniques used for the identification of *Acanthamoeba* and associated intracellular microbes [21,22,33,48,50,55,56,61,72,74,76]. Two studies used gas–liquid chromatography to detect cellular fatty acids of intracellular bacteria and the identification was performed using Microbial Identification Inc. protocols (MIDI) (Newark, DE, USA) [44,46]. PCR (33/43, 76.7%), gene sequencing (30/43, 69.8%), and microscopy (transmission and scanning electron microscopy, confocal laser scanning, and phase-contrast microscopy) (29/43, 67.4%) were the most commonly used techniques to identify the amoeba and intracellular microbes, followed by fluorescence in situ hybridization (12/43, 27.9%) (Appendix A) [21,49,52,53,62,68,74,77]. Two studies observed intracellular bacteria in *Acanthamoeba* cysts [52,80].

### 2.4. Culture Techniques Used to Isolate and Identify Acanthamoeba

*Acanthamoeba* can be axenically cultured [82], which means a culture in which only a single species is present entirely free from other contaminating organisms, i.e., with no food organisms, or by adding live or dead microbes to stimulate the growth of trophozoites [15,83,84]. Samples (clinical or environmental) are cultured on non-nutrient agar (NNA) covered with bacteria where amoebae graze and move away from the inoculation point in order to recover the symbiont with its natural amoeba host [85]. Axenic culture medium that supports *Acanthamoeba* growth consists of protease peptone, yeast extract, glucose (PYG), and inorganic salts (MgSO_4_ × 7H_2_O, sodium citrate dihydrate × 2H_2_O, Na_2_HPO_4_ × 7H_2_O, KH_2_PO_4_, Fe(NH_4_)_2_(SO_4_)_2_ × 6H_2_O) [86,87]. A wide range of bacteria have been used in co-culture with *Acanthamoeba.* The most common microbes used to culture *Acanthamoeba* are *E. coli*, *Klebsiella aerogenes* [88,89,90] and *Enterobacter* spp. (E. *cloacae* and *E. aerogenes*) [8,25,59] on NNA or in saline [83] (Figure 3). It is not entirely clear why *E. coli* or *K. aerogenes* are the most commonly used as food supplement for culturing *Acanthamoeba* spp. There are only a few studies examining whether Gram negative or Gram positive are preferred or whether bacterial preference is dependent on amoebal species or genotypes [88]. One such study has shown that *Acanthamoeba* grows better on *E. coli, Salmonella enterica* serovar Typhimurium, or *Bacillus subtilis* than *Enterococcus faecalis* or *Staphylococcus aureus* [91].

The bacteria used are commonly heat-killed [86,92] or heat-inactivated [56,62] and spread upon NNA plates [70]. The use of bacteria, even dead bacteria, to grow *Acanthamoeba* trophozoites could potentially affect the types of intracellular microbes that can be grown from the *Acanthamoeba*. Twelve studies have examined the presence of intracellular bacteria using axenic culture [22,43,46,51,66,69,71,72,78,79,80], where three studies [58,71,72] have used antibiotics (streptomycin, penicillin, and gentamicin) in PYG to grow amoebae axenically, 18 studies have used NNA with live/inactivated or killed bacteria (*E. coli, E. cloacae, S. cerevisiae, E. aerogenes*), followed by axenic culture, to recover the intracellular microbes harbouring *Acanthamoeba* [20,21,49,53,56,57,59,61,62,64,65,67,68,75,76,77,81] and antibiotics (penicillin, streptomycin, ampicillin, and amphotericin B) were added in culture media (NNA, TSB, SCGYE, PYG) to make the growth contamination free and axenic in another seven studies [40,42,45,47,48,52,70] (Table 2). Some studies have used PYG without inorganic salts to maintain axenic growth of amoeba [69,72]. In the absence of established method for the recovery and identification of intracellular microbes of amoeba, different methods have been used to cultivate intracellular microorganisms carrying *Acanthamoeba*, which has shown inconsistent results. Pathogenic bacteria, such as *Mycobacterium* spp. [55,66,79] and *Pseudomonas* spp. [72,74,79], were often detected by axenic culture technique, whereas pathogenic intracellular bacteria belonging to the genera *Legionella*, *Pseudomonas*, *Mycobacterium,* and *Chlamydia* in clinical isolates of *Acanthamoeba* were detected by culturing on NNA pre-seeded with heat killed *E. coli* followed by axenic culture in 1X Page’s saline solution [21]. Ten studies have used antibiotics at some point of cultivation to maintain the axenic culture and they have reported limited intracellular microorganisms as compared to studies grown *Acanthamoeba* on NNA supplemented with bacteria, where phylogenetically varied intracellular bacteria were repeatedly detected. In addition, axenic culture has been frequently used for clinical specimens (5/8) and NNA with pre-seeded bacteria was preferred to culture environmental samples (22/26). Four serotypes of Adenovirus (Ad1, Ad2, Ad8, and Ad37) were detected in water-isolated *Acanthamoeba* by growing amoeba in PYG with gentamicin (50 μg/mL) [58].

The co-culture of environmental samples with symbiont-free *Acanthamoeba* as a surrogate host is being used as a new method to grow and recover facultative or obligate intracellular bacteria [93,94,95], but this method is not appropriate for isolating symbiont bacteria together with natural host.

Some bacteria have been examined for their ability to survive co-culture with *Acanthamoeba*. *S. aureus* can grow within *A. polyphaga* strain [91]. *Shigella dysenteriae* and *S. sonnei* were able to survive in co-culture with *A. castellanii* for >3 weeks [96] and mycobacterial strains related to *M. intracellulare* and *M. avium* for six years without any amoebal cytopathic effects [55]. Co-culture of *C. jejuni* with amoebal cells resulted in longer survival times as compared to bacteria grown alone [97]. *C. jejuni* and *L. pneumophila* were able to be resuscitated from a viable-but-nonculturable (VBNC) state when co-cultured with *A. polyphaga* or *A. castellanii*, respectively [97,98]. *Acanthamoeba* co-culture has been used to enrich low bacterial concentrations of four *Campylobacter* species, *C. jejuni*, *C. lari*, *C. coli*, and *C. hyointestinalis* [99]. VBNC *P. aeruginosa* can become culturable and active within 2 h of *Acanthamoeba* ingestion [100]. In vitro studies have shown *A. castellanii* can act as an important environmental reservoir of highly infectious bacteria, such as *Francisella tularensis* and *V. cholerae* [101,102]. Furthermore, *V. cholerae* survives within the contractile vacuole of amoeba, even upon the encystment and *F. tularensis* grows faster in co-culture with amoeba when compared to bacteria grown alone and causes rapid amoebal encystment [103]. Similarly, viable and intact growth of *Helicobacter pylori* is increased when co-cultured with *A. castellanii* [104]. Spores of a virulent *B. anthracis* (Ames strain with both pX01 and pX02 virulence plasmids, and Sterne strain with only pX01), an agent of bioterrorism, have shown a 50-times increase in spore count after 72 h of co-culture with *A. castellanii*. In addition, the spores were germinated within phagosomes of amoeba, with the Sterne strain showing less growth [105]. Pathogenic bacteria, such as *A. baumannii*, *K. pneumoniae,* and *E. coli* have been recovered from water samples by *A. polyphaga* co-culture [93]. *Acanthamoeba* also promotes the survival and growth of fungi and viruses (Table 3), suggesting that *Acanthamoeba* can act as an environmental incubator for medically important prokaryotes and fungi.

### 2.5. Species and Genotypes of Acanthamoeba spp.

The most commonly reported genotypes of *Acanthamoeba* are T4 [11,52,57,64,73,74,76,77,81], followed by T3 [58,69,71,72,75,80], T5 [69,72,80,110], and T2 [33,58,62,74] (Appendix A). *A. polyphaga* was detected in eight studies [20,33,47,53,54,65,70,78] and *A. castellani* was observed in five studies [20,33,56,66,70]. *A. hatchetti* (T11, T4) [20,22] and *A. palestinensis* (T2, T6) [20,67] were observed in two studies. Additionally, *A. culbertsoni*, *A. astronyxis* (T7) [20], *A. lugdunesis* [41], *A. mauritaniensis* [42], and *Acanthamoeba* T7 [58] and T11 [75] strains were also reported by single studies.

### 2.6. The Types of Microorganisms Commonly Found Inside Acanthamoeba spp.

Bacteria were the most commonly identified intracellular microorganism in *Acanthamoeba* followed by viruses and fungi (Appendix A). Unidentified bacteria, termed *Candidatus,* were reported in 1/3^rd^ of included studies [22,33,42,47,50,51,52,53,57,61,65,69,72,77,78,80] and *Chlamydia* species were detected in 11 studies [21,33,42,49,57,62,64,69,70,76,78]. Five studies found *Legionella* spp. [21,44,64,75,79], another five studies reported *Mycobacterium* [21,55,66,78,79] or *Pseudomonas* spp. [21,71,74,79,81], four studies found *Rickettsia* spp. [48,77,78,111], three studies detected *Cytophaga* spp. [46,52,56], and *E. coli* [73,81] or *Stenotrophomonas maltophilia* [68,81] were detected in two studies. *Burkholderia pickettii* [43], *Agrobacterium tumefaciens* [74], *Brevibacillus* sp. [81], *Flavobacterium* sp. [52], *Brevundimonas vesicularis,* or *Microbacterium* sp. [81] were also reported in single studies. Three studies only reported the morphology of intracellular “bacteria” present in *Acanthamoeba* [20,40,41]. An archaea-like organism was detected in the cytoplasm of *Acanthamoeba* recovered from a potable water reservoir [45].

Giant mimivirus was detected in three studies [54,65,67], and human adenovirus (HAdV) was isolated in two studies [58,81]. The virophage sputnik 2 [65] and pandoravirus [59] were detected in the contact lens of AK patient in one study. *Aspergillus* was found in *Acanthamoeba* recovered from corneal scrapes and contact lenses of a keratitis patient in one study [81].

The presence of more than one intracellular microbe was reported in ten studies [21,22,57,62,65,70,71,78,79,81]. For example, *Chlamydia* and *Legionella* have been observed in a clinical isolate of *Acanthamoeba,* an environmental isolate that harboured *Legionella* and *Mycobacterium* [21], and *Procabacter* and *Parachlamydia* were found in *Acanthamoeba* (OEW1) isolated from a saline lake in Austria [57]. A study from Iran reported three intracellular microorganisms, *P. aeruginosa*, *Aspergillus* spp. and HAdV in a clinical isolate of *Acanthamoeba* T4 (ICS7) [81]. *A. polyphaga* isolated from a keratitis patient hosted four intracellular prokaryotes: Deltaproteobacterium, Alphaproteobacterium, mimivirus Lentille, and the virophage Sputnik 2 [65].

### 2.7. Differences between the Intracellular Prokaryotes Found in Environmental and Clinical Isolates of Acanthamoeba

Twenty-six studies (60.5%) analysed environmental samples that were collected from soil, sewage sludge, water treatment plants, household tap water, recreational water sources, air conditioning units, hospital areas, such as operating theatres, and contact lens storage cases. Eight (18.6%) studies processed specimens from patients, such as nasal or mucosal swabs, corneal scrapes/swabs or tissue, and AK patient’s contact lenses, and these were grouped as clinical samples. Another nine studies (20.9%) examined both types of samples (Appendix A and Table 1).

Pathogenic bacteria, such as *E. coli*, *Mycobacterium* spp. and *P. aeruginosa,* were observed in *Acanthamoeba* strains that were cultured from clinical specimens [21,66,73,81] (Table 4). *Acanthamoeba* spp. obtained from the corneas of patients contained obligate intracellular bacteria of the order Rickettsiales [48,111], *E. coli* [73], *Pseudomonas*, *Chlamydia* [21], *Caedibacter caryophilus* and *Cytophaga-Flavobacterium-Bacteroides* (CFB) [56]. The presence of bacteria in *Acanthamoeba* has been shown to exacerbate keratitis [21,112] and influence the virulence, pathogenicity, and susceptibility of keratitis causing amoeba to therapeutic drugs [55,75]. *Chlamydia* was observed in *Acanthamoeba* isolated from the nasal mucosa of volunteers [42] and presence of *Pandoravirus inopinatum* was confirmed in *Acanthamoeba* strain recovered from pieces of contact lenses worn by a keratitis patient [59,60].

*Acanthamoeba* carrying mimivirus and *Legionella* spp. were isolated from environmental samples that were collected from air-conditioning units, water treatment plants, and sewage sludge [44,54,64,67,75]. Contact lens cases, often cultured when a keratitis case presented for treatment, have been a rich source of intracellular microbes. Mimivirus strain Lentille, Sputnik 2 [65] and *Mycobacterium* sp. [55] have been isolated from contact lens storage cases. Even though contact lens cases are frequently exposed to disinfectants, several studies have shown that these disinfectants often have poor activity against *Acanthamoeba* spp. [113,114,115]. Hospital floor and sink swabs were found to be positive for *Acanthamoeba* with *Chlamydia* (14.3%) showing the possibility of pathogen transmission via amoeba in the hospital setting [76]. Four different serotypes of human adenovirus (HAdV-1, 2, 8, 37) were found in 14.4% (34/236) of amoeba isolated from tap water [58]. *P. aeruginosa* and *A. tumefaciens* were detected in *Acanthamoeba* strains cultured from recreational water samples [74]. *Acanthamoeba* trophozoites and cysts are highly resistant to disinfectants used to decontaminate water supplies and the intracellular bacteria may be protected from these external disinfectants [37,74,116].

Irrespective of the place of isolation, *Acanthamoeba* hosts many different pathogens [18] but endemically important human pathogens, such as *E. coli*, *Pseudomonas* spp. and *Mycobacterium* spp., were more commonly identified in *Acanthamoeba* cultured from clinical specimens, whereas giant viruses (mimivirus and *Pandoravirus*), *Legionella* spp., and unnamed bacteria of genus *Candidatus* were often detected in environmental *Acanthamoeba*. This suggests that most intracellular microbes interact with *Acanthamoeba* in their natural environment [117]. *Acanthamoeba* may act as a “Trojan horse” for microbes, providing them with the opportunity to colonise or infect different environments [118]. The ability of *Acanthamoeba* to host several different intracellular microbes suggests that these may interact with each other and lead to highly complex differences in the pathogenesis of *Acanthamoeba* [21].

## 3. Discussion

This study systematically analysed 43 published studies assessing the reported intracellular microorganisms that were associated with clinical and environmental isolates of *Acanthamoeba*. PCR followed by gene sequencing and microscopy were the most common laboratory techniques used to identify the intracellular microbes. Potentially pathogenic bacteria, such as *Mycobacterium* spp., *P. aeruginosa*, Rickettsiales, and *E. coli*, were often detected in clinical isolates, while *Legionella*, human adenovirus, mimivirus, and uncategorised bacteria (*Candidatus*) were found in environmental isolates. It appeared that the niche from which *Acanthamoeba* had been isolated affected the types of intracellular microbes present, or perhaps affected the ability of particular *Acanthamoeba* strains to cause infections. This latter hypothesis is presented based on previous investigations that domestic water supplies and contact lenses that are exposed to water are risk factors for *Acanthamoeba* keratitis [5,119,120,121]. This suggests that water is the source of the infecting *Acanthamoeba* [122] and, perhaps, those strains that harbour particular intracellular microbes are more able to instigate corneal (or other) infections [21]. However, not all *Acanthamoeba* isolated from infections have been shown to harbour intracellular microbes, perhaps because their presence has not been analysed. Alternatively, the *Acanthamoeba* may expel resident intracellular microbes during the infectious process. These hypotheses require scientific investigation.

NNA with live/heat-inactivated/killed *E. cloacae*/*E. coli* was the most common method (25/43) used for the recovery and identification of *Acanthamoeba* associated microorganisms [21,33,56,68]. A higher proportion of clinical specimens were cultivated using axenic (PYG, NNA, KCM agar) media, while NNA with bacteria was often used to culture environmental samples. Environmental samples may consist of more promiscuous microbes, thus the culture media with *Acanthamoeba* could enhance the recovery and isolation of intracellular bacteria [95]. The use of different bacterial strains to cultivate amoebal trophozoites could affect the intracellular bacteria that can be recovered from the *Acanthamoeba* since different bacteria affect trophozoite growth and encystment [83]. In addition, antibiotics have been used to eliminate live bacteria for the axenic cultivation of *Acanthamoeba*. However, this review supports that use of antibiotics in culture media to grow clinical or environmental *Acanthamoeba* axenically could inhibit amoebal symbionts and limits the recovery of multiple intracellular bacteria. Therefore, before the adaptation to axenic growth, *Acanthamoeba* spp. should be sub-cultured several times on NNA plates that were covered with heat-killed *E. coli* [70], even though *Acanthamoeba* may grow better with live bacteria than heat killed [83]. The use of live *E. coli tolC* knockout mutants on NNA without antibiotics improved the axenic growth of *Acanthamoeba* spp. and these amoebae had phylogenetically distinct intracellular bacteria [70]. There is a definite need to understand whether the food preferences of *Acanthamoeba* depend on its resident sites/species/genotypes or intracellular microbes or change the intracellular community of microbes. Information such as preference for bacterial consumption on growth of amoeba, time for cyst formation, and intracellular survival of bacteria during the cultivation of *Acanthamoeba* have not yet been reported. These dynamics of *Acanthamoeba*-bacteria interaction should be taken into consideration in future studies.

Phylogenetically unrelated intracellular microbes were found within the same isolate of *Acanthamoeba* in ten studies. The diversity of intracellular microbes suggests that their ability to exploit *Acanthamoeba* as a host has developed continually, independent of the phylogenetic lineage [31]. Intracellular microbes can be either in a stable or transient association. Long-term stable interactions have been observed between *Acanthamoeba* and α/β-*Proteobacteria*, chlamydiae, M. *avium* subsp. *paratuberculosis,* and Bacteroidetes [51,52,123]. However, amoeba can release intracellular microbes in suitable environments [124]. Transient association has been reported for bacteria, such as *E. coli* O157:H7, *L. pneumophila*, among others [39,125]. Intracellular survival of enterohaemorrhagic *E. coli* O157:H7 in *A. castellanii* was reduced by Shiga toxins (Stx) that were produced by the bacterium [125]. Co-occurrence of phylogenetically different bacterial species in *Acanthamoeba* can provide an opportunity for lateral gene transfer between intracellular bacteria [57,126]. Multiple-species association within the same host cell poses challenges to all intracellular microbes, such as competition for nutrients obtained from the host cell, while the interplay between intracellular microbes needs to be balanced to ensure the stability of the association [57]. In depth biochemical and genomic analysis are needed in future research to understand the details of the interactions.

Intracellular microbes have been detected in *Acanthamoeba* isolates that belong to genotypes T2–T7, T11, and T13 [33,47,56,58,62,69,75], whether the occurrence of intracellular microbial strains is, in some way, dependent with amoebal genotypes is still an unanswered question. *Acanthamoeba* hosts for a wide range of microbial species that can presumably, and especially if they are permanent residents, resist phagocytosis, survive, multiply, and endure intracellularly [127]. Whether this can train these intracellular bacteria to survive in other cells, such as human macrophages [31,128,129], perhaps by the exchange of genes with other intracellular microbes [130] or by genetic mutation requires further investigation. This hypothesis is further supported by *Chlamydia* species, which use the same strategies to interact with various different host cells and that likely evolved years ago during interaction with primitive unicellular eukaryotes [31]. From a clinical viewpoint, a better understanding of molecular mechanisms by which pathogenic bacteria can resist amoebal phagocytosis may allow for the design of future antibiotics and vaccines in the treatment of intracellular human bacterial pathogens.

## 4. Methods

The Preferred Reporting Items for Systematic reviews and Meta-Analyses (PRISMA) guidelines were followed for this systematic review [131].

### 4.1. Search Strategy and Data Sources

A systematic search was conducted using three electronic databases, PubMed (Medline), Scopus, and Web of Science (WoS), to identify peer-reviewed articles providing information on the types of intracellular microbes associated with *Acanthamoeba* spp. The literature search was performed using the key terms, “Free-living amoeba” OR “FLA” OR “*Acanthamoeba*” AND “Bacterial endosymbiont”/“Bacterial endocytobiont” OR “Intracellular *Acanthamoeba* Endosymbiosis” OR “Amoeba symbiosis” OR “Amoeba-resisting bacteria” as Combinations of Medical Subject Headings (MeSH). This results in searches of articles containing the words ‘*Acanthamoeba*’ AND “Endosymbiont”/“Endocytobiont” OR “*Acanthamoeba* endosymbiosis” OR “Intracellular” OR “Symbiosis” OR “Free-living amoeba” OR “FLA” in their titles and/or abstracts. Additionally, a snow-ball sampling approach was applied while using the reference lists of the selected articles to expand the search. The search was limited to studies that were published in English language and full text articles published between 1 February 1993 to 30 July 2019.

### 4.2. Inclusion Criteria

For an article to be included in this study, it had to be peer-reviewed, available in full text, with its primary objective to isolate and identify intracellular microbes in clinical or environmental isolates of *Acanthamoeba* spp. However, case reports of *Acanthamoeba* with symbionts were included. A narrative review was performed for all of the selected studies.

### 4.3. Exclusion Criteria

Articles that were published in languages other than English, conference abstracts, institutional protocols, other review papers, in vitro studies on the co-culture of *Acanthamoeba* species with bacteria, or other microorganisms for the analysis of symbiosis and isolation of intracellular microbes from amoeba other than *Acanthamoeba* were excluded from the study. Additionally, the coincidental finding of *Acanthamoeba* and microbes in the same sample, but with no evidence of the other microbes being intracellular, were not included in this study.

### 4.4. Data Abstraction, Quality Assessment, and Appraise Risk of Bias in Individual Studies

At first, two members of the review team screened all of the articles, as per the inclusion and eligibility criteria following PRISMA guidelines and excluded inappropriate articles after consultation with the other authors. Following the database search, studies were pooled and uploaded sequentially into EndNote version X9 (Clarivate Analytics, Philadelphia, PA, USA), then duplicate studies were removed from the list. The authors reviewed a selection of the articles to verify the selection methodology. Any discrepancies between the reviewers were resolved by consensus discussion amongst all of the reviewers. Variables of interest in the included studies were laboratory techniques that were used for the identification of microorganisms, detection and types of *Acanthamoeba* and associated intracellular microbial species, study location, type of sample analysed (clinical or environmental), co-occurrence of multiple intracellular microbes within a *Acanthamoeba* cell, and sequence similarity of detected microbes with reference strains.

The potential risk of bias was assessed with a raw score of quality, as per the Newcastle-Ottawa Scale (NOS) guidelines (adapted for cross-sectional and observational studies) for the appropriateness and aims of the study, method of sample collection, and laboratory identification of *Acanthamoeba* and intracellular microbes [132]. A final score was assigned to each study after consensus between the reviewers. NOS scores can vary from 0 to 9, and studies, with an average score of ≥6 were included for this review (Appendix A) [133]. A meta-analysis of the studies was not performed due to a high level of heterogeneity. Therefore, a systematic analysis was performed. Relevant data were extracted from each study in customised datasheets. Because of the diversity in variables in each study, the assessment scale was primarily based on the methodological quality, *Acanthamoeba* identification and evidence of intracellular microbes. Figures were created using Origin Lab, Version 2018 (Northampton, MA, USA).

### 4.5. Outcome Measurements

The main outcome measure of this review was the types of intracellular microbes that were identified dwelling in *Acanthamoeba* species. The secondary outcome measures were the effect of culture techniques on the types of intracellular microbes recovered from *Acanthamoeba* and the type of intracellular microbes from environmental and clinical sources.

## 5. Conclusions

This study systematically reviewed articles on the types of intracellular microorganisms in *Acanthamoeba*. *Acanthamoeba* acts as an incubator and carrier of a wide range of microorganisms. The niche or home of the *Acanthamoeba* appears to affect the types of intracellular microbes. *Chlamydia* spp., *E. coli*, Rickettsiales, *Pseudomonas* spp., and *Mycobacterium* spp. were the most commonly reported microbes in *Acanthamoeba* that were cultured from clinical specimens and *Legionella*, human adenovirus, mimivirus, and bacteria of *Candidatus* group were detected in environmental *Acanthamoeba*. Human macrophage and *Acanthamoeba* share significant cellular and functional features, particularly phagocytic activity, so amoebal cells might train and serve as a preparatory arena for the pathogens to onset diseases in mammalian cells. Molecular-based future studies are expected to assess the microbiome composition residing in *Acanthamoeba* to view the role of amoeba as a universal host and evolutionary trigger of phylogenetically varied microorganisms.

## 6. Limitations of the Study

The major limitation of this review was the lack of meta-analysis due to heterogeneous variables among the included studies. Although the study used multiple search engines using keywords, the query string may not have short-listed all the relevant studies given the disparity in terminology, such as “endosymbiont”, “endocytobiont”, “endosymbiosis”, “amoeba symbiosis”, “intracellular bacteria”, and “amoeba-resisting bacteria”. Additionally, the use of different laboratory techniques to identify the intracellular microbes in the included studies may have biased the reported microbes. Many studies applied protocols to isolate and identify particular prokaryotes, rather than assessing the whole microbiome residing in *Acanthamoeba,* which may not represent all of the microorganisms present within the amoebal cell. This suggests the use of deep sequencing technique could help to identify the composition of amoebal microbiome.

## Figures and Tables

**Figure 1 pathogens-10-00225-f001:**
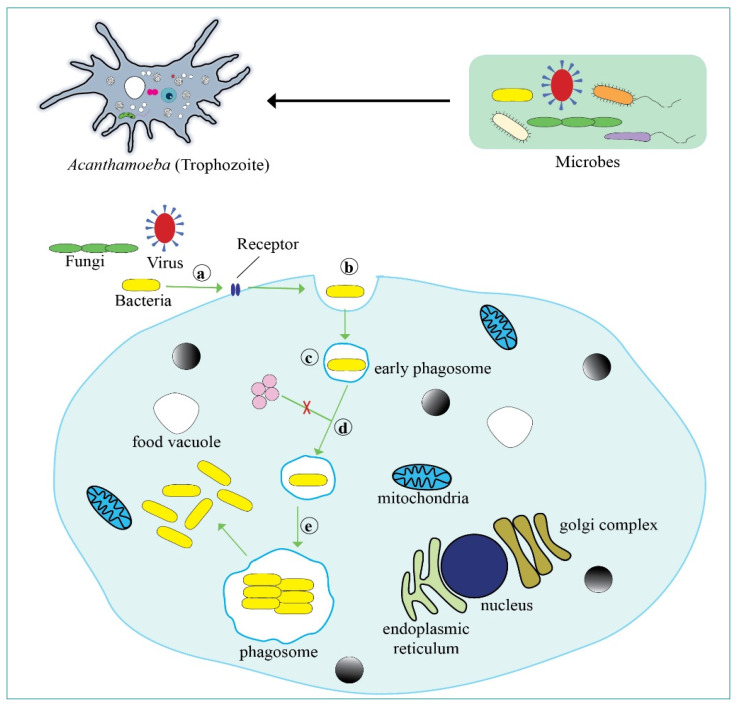
Representation of the different microorganisms as food of *Acanthamoeba* and interaction with bacteria. (**a**) attachment: possible receptor-mediated adhesion of bacteria; (**b**) entry: ingestion of bacteria using pseudopods and phagocytosis; (**c**) trafficking: prevention of phagosome-lysosome fusion by bacteria helps them evade lysosomal degradation and prevents acidification of the phagosomes [39]; (**d**) spread: vacuoles containing microbes disperse throughout the amoebal cytoplasm; and (**e**) replication: intraphagosomal replication of bacteria possible eventual escape into the amoebal cytoplasm.

**Figure 2 pathogens-10-00225-f002:**
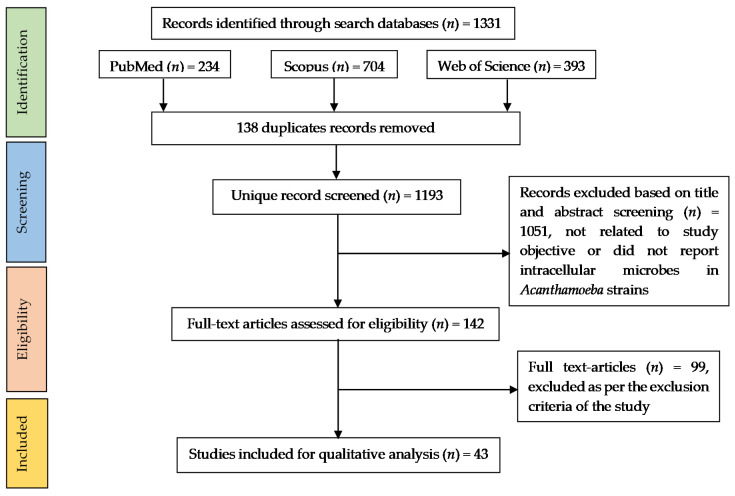
Preferred Reporting Items for Systematic reviews and Meta-Analyses (PRISMA) flow diagram for selection of articles.

**Figure 3 pathogens-10-00225-f003:**
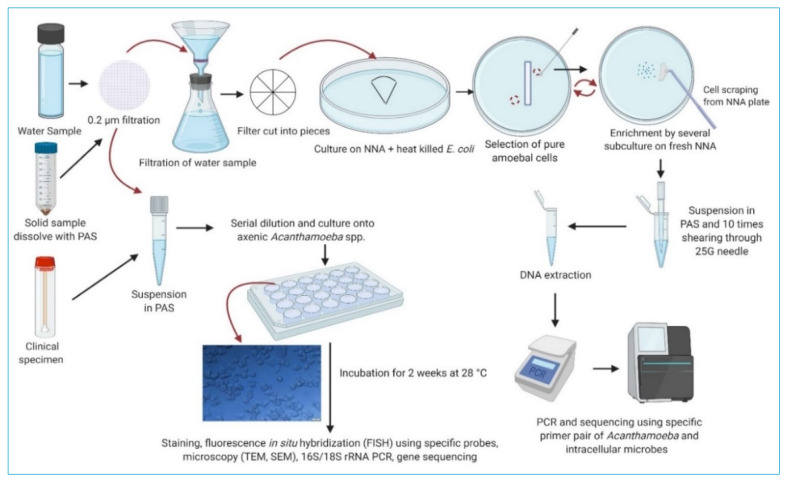
Laboratory procedures for the isolation and identification of *Acanthamoeba* and associated intracellular microorganisms from clinical and environmental samples. Adapted from Thomas et al. (2010) [37]. PAS: Page’s Amoeba Saline, NNA: Non-nutrient agar, TEM = Transmission Electron Microscopy, SEM = Scanning Electron Microscope. Created with BioRender.com (accessed on 20 January 2021).

**Table 1 pathogens-10-00225-t001:** Intracellular microorganisms identified in environmental and clinical isolates of *Acanthamoeba* species.

Country, Date of Study	Analysed Sample (Clinical/Environmental)	Laboratory Investigation	Positive Samples for Intracellular Microbes	Species and Genotypes of *Acanthamoeba*	Identified Intracellular Microbes
USA, 1993 [20]	Clinical (corneal-19, and contact lens-4), environmental specimens (soil, forest detritus, lake and stream sediments, pond water, tree bark, potting soil, 25), and ATCC strains (9)	Culture, electron microscopy, staining	14 of 57	ATCC strains: *A. culbertsoni* 30886, 30011, and 30868 *A. rhysodes* 30973, *A. polyphaga* 30871 and 30461 *A. astronyxis* 30137, *A. hatchetti* 30730, *A. palestinensis* 30870, *Acanthamoeba* strain 30173	Gram-negative rods and cocci and non-acid fast non-motile bacteria
Philippines, 1995 [40]	Pond	Culture, PCR, electronic microscopy	1 of 1	*Acanthamoeba sps*	Gram-negative rod-shaped bacteria, 1.3 × 0.43 µm in size
South Korea, 1997 [41]	Contact lens storage	PCR, TEM	1 of 1	*A. lugdunesis*	Rod-shaped bacteria, 1.38 × 0.5µm in size
Germany, 1997 [42]	Nasal mucosa of humans	Culture, electron microscopy, in situ hybridization	2 of 2	*Acanthamoeba* spp. and *A. mauritaniensis*	Coccoid shaped related to *Chlamydia* spp.; Ca. Parachlamydia acanthamoebae (proposed name for strain Bn9)
Germany, 1997 [43]	Wet area of a physiotherapy unit	Culture, light, and electron microscopy, biochemical tests	1 of 2	*Acanthamoeba* spp. Group II	*Burkholderia pickettii* (biovar 2)
Germany, 1998 [44]	Cold water tap of a hospital plumbing system	Culture, electron microscopy, gas-liquid chromatography	1 of 1	*Acanthamoeba* spp. Group II (K62)	*Legionella*-like slender rods
Germany, 1998 [45]	Potable water reservoir	Culture, electron microscopy	1 of 1	*Acanthamoeba sps* Group II	Archaea like (short rod shaped, 1–1.5 μm length) endoparasite
Germany, 1999 [46]	Drinking water system of a hospital	Culture, phase contrast and electron microscopy, gas-liquid chromatography, Gram staining, biochemical tests	1 of 1	*Acanthamoeba* spp. Group II	*Cytophaga* spp. (K69i)
Germany, 1999 [47]	Two clinical isolates (HN-3 and UWC9) and one environmental isolate (UWE39)	Culture, PCR, Gram and Giemsa staining, sequencing, electron microscopy, FISH, confocal laser scanning microscopy (CLSM)	3 of 3	*Acanthamoeba* spp. (UWC9 and UWE39); *A. polyphaga* (HN-3) [20]	*Ca.* Caedibacter acanthamoebae (proposed name); Ca. Paracaedibacter acanthamoebae (proposed name); Ca. Paracaedibacter symbiosus (proposed name)
USA, 1999 [48]	Corneal scraping	Culture, Gram and Giemsa staining, confocal laser-scanning microscopy, PCR amplification, sequencing of 16S rRNA gene, EM	2 of 2	*Acanthamoeba* species (UWC8 and UWC36)	Phylogenetically related to members of the order Rickettsiales branch of the alpha subdivision of the *Proteobacteria* (99.6% sequence similarity to each other), Ca. Midichloriaceae family in Rickettsiales
USA, 2000 [49]	Clinical (corneal tissues—1), and environmental isolates (soil samples from the USA—1, and sewage sludge from Germany—1)	Culture, Giemsa staining, FISH, electron microscopy, PCR, sequencing	4 of 4	*Acanthamoeba* spp.	Gram-negative cocci, may represent distinct species of Parachlamydiaceae *Ca.* Protochlamydia amoebophila (UWE25) [50]
Greece, 2000 [51]	Water sample collected from the drip-tray of the air conditioning unit of a hospital	Culture, Gimenez Staining, microscopy, PCR, 16S rRNA sequencing	1 of 1	*Acanthamoeba sps*	*Ca.* Odyssella thessalonicensis’ gen. nov., sp. nov. [gram negative, rod, and motile] (proposed name); Note: The phylogenetic position, inferred from comparison of the 16S rRNA gene sequence, is within the α-*Proteobacteria*.
Germany, 2001 [52]	Drinking water in a hospital, corneal scrapings of a keratitis patients (Germany) and eutrophic lake sediment (Malaysia)	Culture, phase contrast and electron microscopy, PCR, 16S rRNA sequencing	3 of 3	*Acanthamoeba* spp. T4	*Flavobacterium succinicans* (99% 16S rRNA sequence similarity) or *Flavobacterium johnsoniae* (98% 16S rRNA sequence similarity); *Cytophaga-Flavobacterium-Bacteroides* (CFB) phylum (<82% 16S rRNA sequence similarity)*;* Ca. Amoebophilus asiaticus (proposed name)
Germany, 2002 [53]	Clinical and environmental isolates from the USA and Malaysia	Culture, Gram, Giemsa and DAPI staining, electron microscopy, FISH, PCR, 16S and 23S rDNA-based sequencing	6 of 6	*A. polyphaga* strain Page 23 and *Acanthamoeba* spp.	Rod-shaped Gram-negative obligate bacterial endosymbionts, related to the β-*Proteobacteria*: Ca. Procabacter acanthamoebae’ gen. nov., sp. nov. (proposed name)
France, 2003 [54]	Water of cooling tower	Gram staining, electronic microscopy, genome sequencing	1 of 1	*A. polyphaga*	Mimivirus
South Korea, 2007 [55]	Contact lens storage case	Culture, MtDNA RFLP analysis, TEM, PCR, sequencing, AFB, and fluorescent staining	1 of 1	*A. lugdunensis*	*Mycobacterium* spp.
South Korea, 2007 [56]	From the infected corneas of Korean patients	Culture, orcein staining, RFLP, TEM, PCR, sequence analysis of 16S rDNA of endosymbiontsand 18S rDNA of *Acanthamoeba*	4 of 4	Strains of *Acanthamoeba* spp. belonging to the *A. castellanii* complex T4	*Caedibacter caryophilus* (proposed name); *Cytophaga-Flavobacterium-Bacteroides* (CFB) phylum
Austria, 2007 [57]	Lake	Culture, FISH, TEM, PCR, 16S rRNA sequences	1 of 1	*Acanthamoeba sps* T4	*Ca.* procabacter sp. OEW1 (proposed name); *Parachlamydia acanthamoebae* Bn9
Spain, 2007 [58]	Tap water samples	Culture, PCR	34 of 236	*Acanthamoeba* spp. T2; T3; T4; T6 and T7	Human adenoviruses (HadV); serotypes HadV-1, 2, 8, and 37
Germany, 2008 [59]	Contact lens and storage case fluid	Culture, light and electron microscopy	1 of 1	1. *A. triangularis* 2. Not yet determined, with polygonal cysts	*Pandoravirus inopinatum* [60]
Austria, 2008 [33]	Soil and lake sediment samples from Austria, Tunisia, and Dominica (N=10)	Culture, TEM and confocal laser scanning microscopy, PCR, genotyping, sequencing	8 of 10	*Acanthamoeba* spp. (isolates EI1, EI2, EI3, 5a2, EIDS3, and EI6) = T4 and (isolates EI4 and EI5) = T2	*Parachlamydia* sp. isolate Hall’s coccus; *Protochlamydia amoebophila* UWE25; Ca. Paracaedibacter acanthamoebae (proposed name); Ca. Amoebophilus asiaticus TUMSJ-321 (proposed name); Ca. Procabacter acanthamoebae Page23 (proposed name); *Parachlamydia* sp. isolate UV-7
South Korea, 2009 [61]	Tap water	Culture, TEM and phase-contrast light microscopy, PCR, 16S r DNA sequencing	5 of 17	*Acanthamoeba* spp.	*Ca.* Amoebophilus asiaticus (proposed name); Ca. Odyssella thessalonicensis (α-*Proteobacteria*) (proposed name); *Methylophilus* spp.
Japan, 2010 [62]	Environmental samples (41 soil samples: 19 river water samples, 4 lake water samples and 2 pond water samples)	Culture, PCR, sequencing, FISH, TEM	5 of 41	*Acanthamoeba* spp. T2;T4; T6 and T13	Rod-shaped belonging to α- and β-*Proteobacteria* phyla; sphere/crescent-shaped belonging to the order chlamydiales *Protochlamydia; Neochlamydia* [63]
USA, 2010 [21]	*Acanthamoeba* isolates (N=37) recovered from the cornea and contact lens paraphernalia of 23 patients, 1 environmental (water) isolate	Culture, PCR, sequencing, FISH, TEM	22 of 38	*Acanthamoeba* spp.	*Legionella* sp.; *Pseudomonas* sp.; *Mycobacterium* sp.; *Chlamydia* sp.
Spain, 2010 [64]	Three different water treatment plants	Axenic culture, sequencing a portion of the 18S rRNA gene for amoeba and specific 16S rRNA gene PCR for endosymbionts	5 of 9	*Acanthamoeba* T4 strain	Chlamydiae; Legionellae
France, 2011 [65]	Corneal scraping of AK patient, contact lens storage case liquid	Culture, slit-lamp examination, PCR, sequencing, matrix-assisted laser desorption ionization time-of-flight mass spectrometry	1 of 1	*A. polyphaga*	*Ca.* Babela massiliensis/ Deltaproteobacterium (proposed name); *Alphaproteobacterium* bacillus; mimivirus strain Lentille; virophage Sputnik 2
USA, 2011 [66]	Eye infection, *A. castellanii* strain Ma (ATCC 50370), culture collection	Culture, light microscopy, PCR, sequencing	1 of 1	*A. castellanii* (ATCC 50370)	Species of *Mycobacterium avium* complex (MAC) (*M. timonense*; *M. marseillense* and *M. chimaera*).
UK, 2011 [67]	Sewage sludge	Culture, PCR, sequencing of Amoeba only	1 of 1	*A. palestinensis* (22/25 clones) within the T6 clade	Mimivirus-like particles
Germany, 2013 [68]	From biofilm of a flushing cistern in a lavatory	Culture, PCR, sequencing, electron microscopy	1 of 1	*Acanthamoeba* spp.	*Stenotrophomonas maltophilia* complex (96.5% sequence similarity)
Japan, 2014 [69]	Hot Spring in Japan	Culture, FISH, TEM, confocal laser and phase-contrast microscopy, PCR, sequencing	1 of 1	*Acanthamoeba* spp. T5	*Protochlamydia*
Austria, 2014 [70]	Three environmental samples	Axenic culture, PCR, FISH, sequencing	7 of 10	*Acanthamoeba* spp. (closely related to *A. castellanii* Neff GenBank Acc. U07416, *A. polyphaga*)	*Paraceadibacter*; *Neochlamydia*; *Protochlamydia*; *Procabacter*; Rickettsiales; *Amoebophilus*
Brazil, 2015 [71]	Seven samples from air-condition units, and five from contact lens cases	Culture, FISH, semi nested-PCR, DGGE, sequencing	3 of 12	*Acanthamoeba* spp. T3; T4 and T5	*Paenibacillus* spp., Ca. *Protochlamydia amoebophila,* (uncultured γ-*Proteobacterium*) (prposed name)
Brazil, 2015 [72]	Seven samples from air-condition units, and five from contact lens cases	Axenic culture, conventional PCR, amplicon sequencing	12 of 12	*Acanthamoeba* spp. T3; T4 and T5	*Pseudomonas* spp.
Japan, 2015 [73]	Isolated from a patient with AK	Culture, Gram staining, MicroScan autoSCAN-4 system, PCR	1 of 1	*Acanthamoeba* strain T4	*E. coli*
Iran, 2015 [74]	Recreational water sources	Axenic culture, staining, PCR, genotyping, microscopy	5 of 16	*Acanthamoeba* spp. T4 and T5	*P. aeruginosa; Agrobacterium tumefaciens*
Spain, 2015 [75]	Seventy water samples (three DWTP, three wastewater treatment plants and five natural pools)	Culture, PCR, genotyping, sequencing	43 of 54	*Acanthamoeba* T3, T4 and T11	*Legionella* spp.
Japan, 2016 [76]	Smear samples from University Hospital	Culture, PCR, sequencing	3 of 21	*Acanthamoeba* spp. T4	*Protochlamydia* spp.; *Neochlamydia* spp.
Austria, 2016 [22]	Corneal scraping of AK patient	Axenic culture, PCR, sequencing, FISH, TEM	1 of 1	*A. hatchetti*, T4	*Parachlamydia* acanthamoebae; *Candidatus* Paracaedibacter acanthamoebae (proposed name)
Austria, 2016 [77]	Seventy-eight water samples (66 cooling tower water: 2 cooling towers of hospital, 1 cooling tower of company, and 12 tap water)	Culture, FISH, real-time PCR, genotyping, and sequencing	3 of 53	*Acanthamoeba* spp. T4	*Paracaedibacter acanthamoebae*; Rickettsiales; *L. pneumophila*
Canada, 2017 [78]	Five clinical isolates (human cornea, nasal swab, monkey kidney tissue Culture) and four environmental isolates (lake sediment, soil, and water reservoir); all ATCC strains	Axenic culture, amplifying and sequencing of bacterial 16S DNA	3 of 9	*A. polyphaga* ATCC 30173 and 50495; *Acanthamoeba* spp. PRA-220	Holosporaceae (Rickettsiales); *Mycobacterium* spp.; *Parachlamydia* spp.; Ca. *procabacter* sp. (proposed name)
Malaysia, 2017 [79]	Isolates from air-conditioning outlets in wards and operating theatres (dust particles)	Axenic culture, PCR, genotyping, sequencing	29 of 36	*Acanthamoeba* spp.	*Mycobacterium* spp. (*M. fortuitum, M. massiliense, M. abscessus, M. vanbaalenii, M. senegalense, M. trivial and M. vaccae*); *Legionella* spp. (*L. longbeachae, L. wadwaorthii, L. monrovica, L. massiliensis and L. feeleii*); *Pseudomonas* spp. (*P. stutzeri; P. aeruginosa; P. denitrificans; P. chlororaphis* and *P. knackmussi*)
Malaysia, 2018 [80]	Air-condition (11 isolates), and keratitis isolates (2)	Axenic culture, PCR, sequencing, FISH (double), TEM	6 of 13	*Acanthamoeba* spp. T3; T4 and T5	*Ca.* Caedibacter acanthamoebae/*Ca.* Paracaedimonas acanthamoeba and Ca. Jidaibacter acanthamoeba (proposed name)
Iran, 2019 [81]	Corneal scrapes and contact lenses isolate of keratitis patients	Culture, light microscopy, gram staining, PCR, sequencing	7 of 15	*Acanthamoeba* spp. T4	*E. coli*; *Achromobacter sps*; *P. aeruginosa; Aspergillus* sp.; *Mastadenovirus* sp.; *Microbacterium* sp.; *Stenotrophomonas maltophilia*; *Brevundimonas vesicularis* and *Brevibacillus* sp.

**Key:** AFB = Acid Fast Bacilli, ATCC = American Type Culture Collection, AK = *Acanthamoeba* keratitis, PCR = Polymerase Chain Reaction, TEM = Transmission Electron Microscopy, SEM = Scanning Electron Microscope, FISH =Fluorescence in situ Hybridization, DWTP = Drinking Water Treatment Plant, DAPI = 4′,6-diamidino-2-phenylindole, MtDNA = Mitochondrial DNA, RFLP = Restriction Fragment Length Polymorphism, DGGE = Denaturing Gradient Gel Electrophoresis, Ca. = *Candidatus.*

**Table 2 pathogens-10-00225-t002:** The types of microbes isolated from *Acanthamoeba* spp. using different culturing techniques.

Culture Type	Source of *Acanthamoeba*	Identified Intracellular Organism in *Acanthamoeba*	Study
Axenic culture on PYG, KCM agar, NNA (*n*= 12)	Clinical isolates	*Mycobacterium avium* complex	[66]
		*Escherichia coli*	[73]
		*Parachlamydia* acanthamoebae and Ca. Paracaedibacter acanthamoebae	[22]
	Environmental isolates	*Candidatus* spp.	[51]
		*Protochlamydia*	[69]
		*Burkholderia pickettii* (biovar 2)	[43]
		*Cytophaga* spp.	[46]
		*Mycobacterium* spp.	[55]
		*P. aeruginosa* and *Agrobacterium tumefaciens*	[74]
		*Mycobacterium* spp. and *Pseudomonas* spp.	[79]
	Clinical and environmental (both) isolates	Rickettsiales; *Mycobacterium* spp.; *Parachlamydia* spp. and Ca. procabacter sp.	[78]
		*Candidatus* spp.	[80]
Axenic culture in presence of antibiotics (*n* = 3)	Environmental isolates	Human adenoviruses	[58]
		*Paenibacillus* spp.; Ca. *Protochlamydia amoebophila;* γ-*Proteobacterium*	[71]
		*Pseudomonas* spp.	[72]
NNA with live/inactivated/killed bacteria (*n*= 18)	Clinical isolates	*E. coli; Achromobacter sps; P. aeruginosa; Aspergillus sps;* Mastadenovirus; *Microbacterium sps; Stenotrophomonas maltophilia; Brevibacillus sps* and *Brevundimonas vesicularis*	[81]
		*Caedibacter caryophilus* and *Cytophaga-Flavobacterium-Bacteroides*	[56]
	Environmental isolates	*Ca.* Babela massiliensis, *Alphaproteobacterium* bacillus, Mimivirus (Lentille), Virophage (Sputnik 2)	[65]
		Mimivirus-like particles	[67]
		*Stenotrophomonas maltophilia* complex	[68]
		*Legionella* spp.	[75]
		*Ca. procabacter* sp. and *Parachlamydia acanthamoebae*	[57]
		*Protochlamydia* spp. and *Neochlamydia* spp.	[76]
		*Paracaedibacter acanthamoebae;* Rickettsiales; *L. pneumophila*	[77]
		Pandoravirus	[59]
		*Parachlamydia* sp.; *Protochlamydia amoebophila*; *Candidatus* spp.	[33]
		*Candidatus* spp.	[61]
		α- and β-*Proteobacteria* and chlamydiales	[62]
		Chlamydiae; Legionellae	[64]
	Clinical and environmental (both) isolates	Gram-negative; rods and coccus; non-acid fast; non-motile	[20]
		Parachlamydiaceae and Ca. Protochlamydia amoebophila	[49]
		*Ca.* Procabacter acanthamoebae’ gen. nov., sp. nov. (proposed)	[53]
		*Legionella; Pseudomonas; Mycobacterium; Chlamydia*	[21]
Live/inactivated/killed bacteria on NNA/SCGYE/TSB/PYG with antibiotics (*n*= 7)	Clinical isolates	*Chlamydia* spp. and Ca. Parachlamydia acanthamoebae	[42]
		Rickettsiales spp.	[48]
	Environmental isolates	Archaea like organism	[45]
		Gram-negative, rod-shaped bacteria	[40]
		*Paraceadibacter*; *Neochlamydia*; *Protochlamydia*; *Procabacter*; Rickettsiales; *Amoebophilus*	[70]
	Clinical and environmental (both) isolates	*Candidatus* spp.	[47]
		*Flavobacterium* spp. and Ca. Amoebophilus asiaticus	[52]

**Key:** PYG = Peptone-yeast-glucose, KCM agar = KCM buffer (KCl, CaCl_2_ and MgSO_4_.H_2_O) in Bacto agar, NNA = non-nutrient agar, TSB = Tryptic soy-yeast extract broth, SCGYE = Serum-casein glucose yeast extract.

**Table 3 pathogens-10-00225-t003:** Interactions of fungi or viruses with *Acanthamoeba* spp.

S.N.	Microorganisms	Interaction with *Acanthamoeba* spp.	Reference
1.	**Fungi**		
	*Histoplasma capsulatum*	Co-culture with *A. castellanii* (ATCC 30324), cell lysis	[106]
	*C. neoformans*	Intracellular multiplication in *A. castellanii* strain 30324	[107]
	*Sporothrix schenckii*	Co-culture with *A. castellanii* (ATCC 30324), cell lysis	[106]
	*Fusarium conidia*	Co-culture with different strains of *A. castellanii* (ATCC 30010, 50492), germinate in amoebal cells	[108]
2.	**Viruses**		
	HAdV	Co-culture with different isolates of *Acanthamoeba*, intracellular survival	[58]
	Coxsackie virus	Intracyst and intracellular survival in a clinical isolate of *A. castellanii*	[109]
	Mimivirus	Intracellular multiplication in *A. polyphaga* isolated from the water sample of a cooling tower	[54]

**Table 4 pathogens-10-00225-t004:** Intracellular microbes identified in *Acanthamoeba* spp. from clinical or environmental sources.

Sample Type	Analysed Sample	Amoebal Host	Identified Intracellular Pathogenic Microbes in *Acanthamoeba* spp.	Study
Clinical specimens	Corneal specimens	*Acanthamoeba* spp.	*Legionella*, *Pseudomonas*; *Mycobacterium*; *Chlamydia*	[21]
		*A. castellanii* (ATCC 50370)	*Mycobacterium avium* complex (MAC)	[66]
		*A. polyphaga* (ATCC 50495)	*Mycobacterium* spp.	[78]
		*Acanthamoeba* spp.	Rickettsiales	[48,111]
		*Acanthamoeba* spp. T4	*P. aeruginosa*; *Aspergillus* spp.; Mastadenovirus spp.	[81]
		*A. castellanii* T4	*Caedibacter caryophilus*; *Cytophaga-Flavobacterium-Bacteroides* (CFB)	[56]
		*A. hatchetti* T4	*Parachlamydia* acanthamoebae	[22]
		*Acanthamoeba* T4	*E. coli*	[73,81]
	Human nasal mucosa	*Acanthamoeba* spp.	*Chlamydia sps; Candidatus Parachlamydia* acanthamoebae	[42]
		*A. polyphaga* (ATCC 30173)	Rickettsiales	[78]
	Contact lens and fluid	*Acanthamoeba* spp. (*A. triangularis*)	*Pandoravirus inopinatum*	[59,60]
Environmental samples	Tap water	*Acanthamoeba* (T2, T3, T4, T6, and T7)	Human adenoviruses	[58]
	Recreational water sources	*Acanthamoeba* (T4, T5)	*P. aeruginosa; A. tumefaciens*	[74]
	Water treatment plant, natural pools	*Acanthamoeba* (T3, T4, T11)	*Legionella* spp.	[64,75]
	Sewage sludge and cooling tower water	*A. palestinensis*; *A. polyphaga*	Mimivirus	[54,67]
	Contact lens storage case/liquid	*A. lugdunensis*	*Mycobacterium* spp.	[55]
		*A. polyphaga*	Deltaproteobacterium; Mimivirus Lentille; Virophage Sputnik 2; *Alphaproteobacterium* bacillus	[65]
	Soil and lake sediment	*A. castellanii* and *A. royreba* T4; *A. pustulosa* and *A. polyphaga* T2	*Parachlamydia* sp.; *Protochlamydia amoebophila*; *Ca.* Paracaedibacter acanthamoebae; *Ca.* Amoebophilus asiaticus, *Ca.* Procabacter acanthamoebae	[33]
	Biofilm of a flushing cistern in a lavatory	*Acanthamoeba* spp.	*Stenotrophomonas* spp.	[68]
	Hot Spring	*Acanthamoeba* spp. T5	*Protochlamydia*	[69]
	Hospital environment	*Acanthamoeba* spp. T4	*Protochlamydia* spp.; *Neochlamydia* spp.	[76]
	Tap water	*Acanthamoeba* spp.	*Ca.* Amoebophilus asiaticus; α-*Proteobacteria*; *Methylophilus sps*	[61]
	Recreational water sources	*Acanthamoeba* spp. T4 and T5	*P. aeruginosa* and *Agrobacterium tumefaciens*	[74]
	Lake water	*Acanthamoeba sps* T4	*Parachlamydia acanthamoebae*; *Ca. procabacter* sp.	[57]

## Data Availability

The data presented in this study are available in Table 1, Table 2 and Table 3, Appendix A.

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
