# Peer review of "A Systematic Review of Intracellular Microorganisms within Acanthamoeba to Understand Potential Impact for Infection"

_pathogens, 2021, doi:10.3390/pathogens10020225_

Round 1

Reviewer 1 Report

Rayamajhee et al Review the current knowledge and data of interaction of Acanthamoeba with some pathogenic bacteria. It is nicely written, and from the interesting point of view. However, some important human pathogens and their interaction with amoeba are missing, and should be included, for example Francisella tularensis.

Reviewer 2 Report

Acanthamoeba spp. are ubiquitous facultative pathogens. Acanthamoeba infection present a serious risk to human health and are characterized by high mortality, especially in immunocompromised individuals. Moreover, Acanthamoeba spp. have been working as an excellent vector for pathogens, as a "Trojan horse". Studies have demonstrated that the interaction of pathogens with Acanthamoeba spp. results in induction and maintenance of virulence factors and increase in microbial pathogenicity. Therefore, the research conducted by Rayamajhee B, Subedi D, Peguda HK, Willcox M, Henriquez FL and Carnt N is very imporant in the field. Some minor points could, however, been improved:

1) 1. Introduction: please change "eye" into "cornea" in the line 40. Moreover, I don't understand what authors had in mind writing "nasal passages and the respiratory track". Amoebas can entry through the nasal passages to the lower respiratory tract.

2) 2.1.  Results of the search: The flowchart should be named as Figure 2. Moreover, the flowchart is not readable (maybe due to conversion of the file into pdf). Please check it and/or change it.

3) 2.2. Included studies. Table 1 should be under the paragraph 2.2. It would be easier to read the text. Now, it's on the page 7, and the reference to the Table 1 is on page 4.

4) 2.3. Laboratory techniques ...: Why in the text is Figure B? Where is Figure A? Please change it.

5) 2.4. Culture techniques..: In the line 148, NNA is mentioned for the first time, and in this place the abbreviation should be explained. In the line 148, "NNA" change into "non-nutrint agar (NNA)", while in the line 155, please delete "non-nutrient agar"

6) 2.4. Culture techniques..: I would suggest to put Figure 2 (which is excellent) under the text, where it's mentioned (in the line 161)

7). 2.4. Culture techniques..: In the line 172, you mentioned Table 3 when you haven't mentioned Table 2 yet. I'm confused. Number the tables in the order they appear in the text. Tables should be under the text, where they are mentioned. It's more legible.

8) 2.5. Genotypes of Acanthamoeba spp.: The paragraph is called genotypes, and the text is about the species of Acanthamoeba (A. castellani, A. polyphaga). Change the name of the paragraph or the text. Moreover, I have objections about Figure C. You mixed species and genotypes in the figure, where for example: A. castellani, A. polyphaga, A. mauritaniensis are also T4. I suggest to present in the figure either only genotypes or only species.
